The herbaceous landlord: integrating the effects of symbiont consortia within a single host

Vandegrift Roo 1 awv@uoregon.edu
Roy Bitty A. 1
Pfeifer-Meister Laurel 1 2
Johnson Bart R. 3
Bridgham Scott D. 1 2
1 Institute of Ecology and Evolution, University of Oregon , Eugene, OR , United States
2 Environmental Science Institute, University of Oregon , Eugene, OR , United States
3 Department of Landscape Architecture, University of Oregon , Eugene, OR , United States
Martin Francis
Electronic publication date: 2015 Nov 3
Publication date: 2015
Volume: 3
Electronic Location ID: e1379
Received 2015 Jul 7; Accepted 2015 Oct 14
Copyright: © 2015 Vandegrift et al.
Copyright year: 2015
Copyright holder: Vandegrift et al.
License: This is an open access article distributed under the terms of the Creative Commons Attribution License, which permits unrestricted use, distribution, reproduction and adaptation in any medium and for any purpose provided that it is properly attributed. For attribution, the original author(s), title, publication source (PeerJ) and either DOI or URL of the article must be cited.
License URL: https://creativecommons.org/licenses/by/4.0/

Keywords: Epichloë, Agrostis capillaris, Symbiosis, Mycology, Climate change, Prairies, Systems ecology, Mutualist-pathogen continuum, AMF, DSE

Funding: Office of Science (Biological and Environmental Research) US Department of Energy DE-FG02-09ER604719 National Science Foundation (NSF) MacroSystems Biology Program, Award 134087 NSF Graduate Research Fellowship (DGE-0829517) This research was funded by the Office of Science (Biological and Environmental Research), US Department of Energy, grant number DE-FG02-09ER604719, and the National Science Foundation (NSF), MacroSystems Biology Program, Award Number 134087. Roo Vandegrift was supported by an NSF Graduate Research Fellowship (DGE-0829517). The funders had no role in study design, data collection and analysis, decision to publish, or preparation of the manuscript.

==============================
Plants are typically infected by a consortium of internal fungal associates, including endophytes in their leaves, as well as arbuscular mycorrhizal fungi (AMF) and dark septate endophytes (DSE) in their roots. It is logical that these organisms will interact with each other and the abiotic environment in addition to their host, but there has been little work to date examining the interactions of multiple symbionts within single plant hosts, or how the relationships among symbionts and their host change across environmental conditions. We examined the grass Agrostis capillaris in the context of a climate manipulation experiment in prairies in the Pacific Northwest, USA. Each plant was tested for presence of foliar endophytes in the genus Epichloë, and we measured percent root length colonized (PRLC) by AMF and DSE. We hypothesized that the symbionts in our system would be in competition for host resources, that the outcome of that competition could be driven by the benefit to the host, and that the host plants would be able to allocate carbon to the symbionts in such a way as to maximize fitness benefit within a particular environmental context. We found a correlation between DSE and AMF PRLC across climatic conditions; we also found a fitness cost to increasing DSE colonization, which was negated by presence of Epichloë endophytes. These results suggest that selective pressure on the host is likely to favor host/symbiont relationships that structure the community of symbionts in the most beneficial way possible for the host, not necessarily favoring the individual symbiont that is most beneficial to the host in isolation. These results highlight the need for a more integrative, systems approach to the study of host/symbiont consortia.

Introduction

There has been a surge in interest in the microbiome of terrestrial plants (Porras-Alfaro & Bayman, 2011; Turner, James & Poole, 2013), largely driven by the increasing recognition that the microbial associates of plants play major roles in plant health (Carroll, 1988; Chaparro et al., 2012; Berendsen, Pieterse & Bakker, 2012; Berlec, 2012). Furthermore, microbial associates of plants may be integral to plants’ responses to disease and climate change (Köberl et al., 2011; Woodward et al., 2012). Here we ask: how do the associations of microbes change under different climatic conditions within the same host plant species, and does this matter to host fitness?

Particularly important components of the plant microbiome are fungal symbionts, especially mycorrhizal fungi (Munkvold et al., 2004; Glassman et al., 2015) and fungal endophytes (Arnold & Lutzoni, 2007; Porras-Alfaro & Bayman, 2011) (Box 1). Fungal endophytes are defined functionally, rather than phylogenetically—they are fungi found within living, healthy plant tissues (Clay, 1990; Rudgers et al., 2009). Endophytes make their living by not harming their host enough to induce a defensive reaction; many of these fungi are assumed to be mutualists, but both fungal endophytes and mycorrhizal fungi exist on a functional continuum from mutualist to pathogen (Carroll, 1988; Porras-Alfaro & Bayman, 2011). The position upon this continuum will depend upon the environmental context, in addition to the particular host/symbiont pairing (Carroll, 1988; Johnson, Graham & Smith, 1997; Saikkonen et al., 1998; Saikkonen et al., 2006; Faeth & Sullivan, 2003).

While there is a growing body of research examining the interactions among multiple symbionts within a single host (Müller, 2003; Lingfei, Anna & Zhiwei, 2005; Novas, Cabral & Godeas, 2005; Novas et al., 2011; Omacini et al., 2006; Mack & Rudgers, 2008; Scervino et al., 2009; Kandalepas et al., 2010; Urcelay, Acho & Joffre, 2011; Liu et al., 2011), most studies of fungal symbionts of plants have examined individual relationships in isolation (Kuldau & Bacon, 2008; Porras-Alfaro & Bayman, 2011; White & Bacon, 2012), despite recognized need for an integrative, systems biology perspective (Porras-Alfaro & Bayman, 2011; Schlaeppi & Bulgarelli, 2015). In this study we examined the interaction of three symbionts within a single grass host, as well as the shift in host/symbiont interactions within the context of a manipulative climate change experiment.

We focus on three groups of symbionts (Box 2). Fungi in the genus Epichloë are endophytes that systemically infect the aboveground tissues of many grasses (Fig. 1A), and are often assumed to be strong mutualists (Schardl, 1996; Bush, Wilkinson & Schardl, 1997; Scott, 2001), though they may also be pathogenic (Faeth & Fagan, 2002; Brem & Leuchtmann, 2002; Vandegrift et al., 2015a). Arbuscular mycorrhizal fungi (AMF; Fig. 1B) colonize the roots of the vast majority of terrestrial plants (∼80% of plant families) (Schüßler, Schwarzott & Walker, 2001) and provide access to inorganic soil nutrients in exchange for photosynthate (Harley & Smith, 1985). Dark septate endophytes (DSE; Fig. 1C) are a poorly studied, phylogenetically diverse group of root-inhabiting fungal endophytes (Jumpponen, 2001; Porras-Alfaro & Bayman, 2011). Though previously assumed to be pathogens (Jumpponen & Trappe, 1998), there is mounting evidence that DSE may function as pseudo-mycorrhizae in some contexts (Upson, Read & Newsham, 2009; Alberton, Kuyper & Summerbell, 2010). All three of these groups of symbionts may exist across the full spectrum of the mutualist/pathogen continuum.

Figure 1 Symbionts and locations within the host plant.

(A) Epichloë endophytes, pictured in red, systemically infect the aboveground tissues of host grasses, growing between cells; (B) AMF colonize the roots of their hosts, forming characteristic nutrient exchange structures called arbuscules (Arb.) and storage vesicles (Ves.); (C) DSE colonize roots as well, and are often found in association with AMF. The photomicrograph (400×) shows brown DSE colonizing the same segment of an Agrostis capillaris root as AMF, with DSE haustoria (Hst.) in close proximity to AMF arbuscules (Arb.).

There is evidence of competition between Epichloë endophytes and AMF in multiple grass species (Brachypodium sylvatica, Lolium perenne, Lolium multiflorum, and Schedonorus phoenix) (Müller, 2003; Omacini et al., 2006; Mack & Rudgers, 2008; Liu et al., 2011). There is also some evidence of a more cooperative relationship in some cases (Novas, Cabral & Godeas, 2005; Novas et al., 2011). It is reasonable to expect that these two types of fungi may interact within all their hosts. Though there is little research on the subject to date, there are some reports suggesting AMF/DSE competition (Kandalepas et al., 2010; Urcelay, Acho & Joffre, 2011), as well as potential facilitation (Lingfei, Anna & Zhiwei, 2005; Scervino et al., 2009).

To examine these multi-symbiont interactions, we quantified percent root length colonized (PRLC) by both AMF and DSE, and tested for the presence of systemic foliar Epichloë endophytes within a single host species (Agrostis capillaris L.) across a broad climatic gradient within the context of a manipulative climate change experiment (detailed in Pfeifer-Meister et al., 2013). We examined how these fungal symbionts interacted to affect host fitness across a broad range of environmental conditions.

Considering that plant symbionts largely exist on a mutualist/pathogen continuum, we hypothesized that we would find evidence that the symbionts in our system were in competition for host resources—photosynthate, space within roots, etc.—and that the outcome of that competition could be driven by the benefit to the host. In other words, we hypothesized that the host plants would be able to allocate carbon to the symbionts in such a way as to maximize fitness benefit within a particular environmental context. Specifically, we expected changes along soil nutrient and soil moisture gradients to alter the balance between symbionts, favoring AMF over DSE and Epichloë in drier and more nutrient-poor soils. We also expected that symbionts could alter host response to environmental conditions; in particular, we expected that correlations between AMF/DSE PRLC and fitness would change predictably along environmental gradients.

Box 1 Definitions of terms

Symbiosis : We use the word symbiosis in the literal sense, meaning “to live together”, for the relationship between a host and an associated fungus. The symbiont is the fungal partner, deriving nutrition from the host; the words symbiont and symbiosis are not intended to convey any sense of whether or not the association is beneficial or harmful to the host, only that the association exists.

Mutualist : A mutualist is a symbiont that provides a net fitness benefit to its host. Mutualism implies that both partners benefit—we take the nutritional mode of the fungal partner (i.e., carbon derived from the host) to be the symbiont’s benefit. For example, some Epichloë endophytes of grasses are mutualists, because they produce fungal alkaloids that can lead to a dramatic reduction in herbivory of the host (Brem & Leuchtmann, 2001; Kuldau & Bacon, 2008; Gange et al., 2012).

Pathogen : We define a pathogen as a symbiont that causes a net fitness decrease in its host. There are many obvious and direct plant pathogens, such as ergot (Claviceps purpurea (Fr.) Tul.), which reduces host fitness by forming sclerotia on the developing seeds of its host (Langdon, 1954). There are, however, many much less direct modes of pathogenicity: some Epichloë endophytes, for example, have been shown to reduce growth rates and seedling survival (Brem & Leuchtmann, 2002; Vandegrift et al., 2015a)—if these fitness costs of hosting the fungus are not offset by fitness benefits provided by the fungus, the net effect is pathogenic.

Box 2 Overview of symbionts examined

Epichloë (Fig. 1A; anamorphic synonym: Neotyphodium) are a genus of predominantly endophytic fungi in the family Clavicipitaceae. Although many Epichloë species may be seedborne, and thus tightly linked to their host’s fitness (Schardl, 1996), horizontal (contagious) transmission is possible via both sexual (Brem & Leuchtmann, 1999) and asexual (Tadych et al., 2007) means. These are systemic foliar endophytes of cool-season grasses (Poaceae), colonizing the aboveground tissues of their hosts (Schardl, 1996); since these fungi do not colonize root tissues, we presume that interactions with root symbionts are primarily via signaling or competition for host photosynthate. They are generally considered strong mutualists because they produce fungal alkaloids, which can reduce herbivory on the host plant (Schardl, 1996; Bush, Wilkinson & Schardl, 1997; Scott, 2001). Some fungi in this genus have also been linked experimentally with drought tolerance and increased competitive abilities (Malinowski, Belesky & Lewis, 2005). A growing body of work, however, demonstrates that they can be pathogenic in certain circumstances (Faeth et al., 1999; Faeth, 2000; Faeth & Fagan, 2002; Brem & Leuchtmann, 2002; Vandegrift et al., 2015a). The metabolic cost to the plant of hosting an Epichloë endophyte must be balanced by the fitness increase that the endophyte provides.

Arbuscular mycorrhizal fungi (AMF; Fig. 1B) are well known fungal symbionts of plants that provide access to inorganic soil nutrients, most notably phosphorus, in exchange for host photosynthate (Harley & Smith, 1985). AMF have also been linked to uptake of other soil nutrients (Li et al., 2006; Smith & Read, 2008), protection from root pathogens (Newsham, Fitter & Watkinson, 1995; Smith & Read, 2008), and drought tolerance (Ruiz-Lozano, Azcon & Gomez, 1995).

The definitive demonstration that a carbon “marketplace” exists between host plants and AMF (wherein plants can allocate carbon to mycorrhizal partners that provide more phosphorous) did not come until relatively recently (Kiers et al., 2011). The existence of such a marketplace provides a mechanism for the discouragement of cheaters, and demonstrates that plants can control where carbon is allocated over fairly fine spatial scales within their root systems (Selosse & Rousset, 2011; Kiers et al., 2011; Grman, Robinson & Klausmeier, 2012).

This is not to say that AMF cannot be pathogenic in certain contexts. For example, if there is an abundance of available phosphorous, non-mycorrhizal plants perform better than those colonized by AMF (Johnson, 1993; Klironomos, 2003; Johnson et al., 2004; Landis & Fraser, 2008). Environmental conditions determine the benefit of the symbiosis for the plant partner.

Dark septate endophytes (DSE; Fig. 1C) are a poorly studied group of fungal endophytes found in plant roots; they are, however, starting to receive more attention (Collins et al., 2008; Urcelay, Acho & Joffre, 2011; Porras-Alfaro & Bayman, 2011). These common, widely distributed root endophytes are distinguished by their brown cell walls, which are darkly pigmented by fungal melanins. Dark septate root endophytes colonize hosts from across the plant kingdom, and include fungi from multiple phyla, though Ascomycota predominate (Jumpponen, 2001). They are known to co-exist with mycorrhizal fungi within plant roots (Girlanda, Ghignone & Luppi, 2002; Li & Guan, 2007). Previously assumed to often be root pathogens (Jumpponen & Trappe, 1998), DSE have recently been linked to increased plant nutrient uptake, particularly of nitrogen (Upson, Read & Newsham, 2009; Alberton, Kuyper & Summerbell, 2010), and growth (Jumpponen, Mattson & Trappe, 1998; Newsham, 1999; Arnold et al., 2000). As with AMF, these fungi exist upon a continuum—the benefits to the host must outweigh the metabolic costs incurred for these fungi to be truly mutualistic (Mandyam & Jumpponen, 2015).

Methods

Site descriptions

This study was conducted within the framework of a large manipulative climate change experiment in PNW grasslands, described fully in Pfeifer-Meister et al. (2013). We utilized two (of three) experimental sites: one at The Nature Conservancy’s Willow Creek Preserve at the southern end of the Willamette Valley in Eugene, Oregon (44°1′34″N/123°10′56″W), and one at The Nature Conservancy’s Tenalquot Prairie Preserve, managed by the Center for Natural Lands Management, in western Washington (46°55′6″N/122°42′47′W). Willow Creek has mean annual precipitation of 1,201 mm, while Tenalquot Prairie has 1,229 mm; mean annual temperatures at the two sites are are 11.4 °C and 9.8 °C, respectively (Pfeifer-Meister et al., 2013). The soil at Tenalquot Prairie is a gravelly sandy loam Andisol (sandy-skeletal, amorphic-over-isotic, mesic Typic Melanoxerand), whereas the Willow Creek soil is a silty-clay loam Mollisol (very-fine, smetitic, mesic Vertic Haploxeroll).

Prairies and oak savannas historically dominated much of the interior valleys along the Pacific coast from central California to southern British Columbia. The two study sites occupy the Willamette Valley and Puget Lowland Level III ecoregions, respectively (US EPA, 2011). These ecosystems were maintained by drought-season fire, often of anthropogenic origin, which prevented succession to woodland or forest (Boyd, 1986; Walsh et al., 2010; Walsh, Whitlock & Bartlein, 2010). Before Euro-American colonization, 50% the Willamette Valley floor and lower foothills was prairie or savanna (Christy & Alverson, 2011). Presently, however, only 2% of this remains (Baker et al., 2002), and such grasslands are among the most endangered ecosystems in the United States due to fire suppression, land-use change, habitat fragmentation, and invasions by exotic plants and animals (Noss, LaRoe & Scott, 1995).

Climate manipulations and plot measures

All plots were treated with spring and autumn applications of the herbicide glyphosate, followed by mowing and thatch removal. In December 2009 all plots were seeded with the same mixture of 32 native upland prairie graminoids and forbs.

Each site had twenty 3 m diameter circular plots (7.1 m2) fully crossing heat (+3.0 °C) and precipitation (+20%) treatments. Temperature was increased in the experimental plots with six overhead 2000-W infrared heat lamps (Kalglo Electronics, Inc., Bethlehem, PA) angled at 45° to the surface (Kimball et al., 2008). Precipitation intensity was increased by 20% by hand-watering from an on-site rainwater collection system using a gauged hose within two weeks of the most recent rainfall. This led to most of the increased precipitation being applied during the wet season, and very little being applied in the summer, mirroring GCM predictions for the region (Meehl et al., 2007; Mote & Salathé, 2010). All ambient temperature plots had wooden imitation heaters suspended overhead, to control for any effect of shading by the infrared heaters. Precipitation treatments were initiated in the spring of 2010, and heating treatments were initiated by autumn of 2010.

Soil temperature was measured continuously at the center of each plot at 10 cm depth by thermistors (model 107; Campbell Scientific, Logan, UT, USA); volumetric water content (0–30 cm) was measured continuously at the center of each plot by time-domain reflectometry (model CS616-L; Campbell Scientific, Logan, UT, USA). Soil nitrogen and phosphorous availabilities (5–10 cm depth) were measured using anion/cation exchange resin probes (PRS™ Western Ag Innovations Inc., Saskatoon, Canada) from April–July 2011. Nitrogen from ammonium and nitrate ions were combined into a single measure of inorganic nitrogen, though nitrate predominated at both sites.

Focal species and sample collection/preparation

Agrostis capillaris L. (colonial bentgrass) is a perennial bunchgrass native to Eurasia with a stoloniferous habit and an observed preference for dry soils (Hubbard, 1984). Despite this observed preference, reports of its drought tolerance are conflicting (Hubbard, 1984; Dixon, 1986; Ruemmele, 2000). Since our central questions resolved around the interactions between Epichloë, AMF, and DSE within a single host, we chose a grass species that hosts all three symbionts. We focused on an introduced species so that harvesting for our study did not affect the community ecology experiments that were concurrently underway at these sites. Agrostis capillaris plants within the treatment plots were most likely germinants from the seed bank following the herbicide treatments, or potentially germinants from seeds dispersed in from the surrounding fields; there is also a small potential that some stolons survived the herbicide treatment.

In June–July of 2011 we collected four first-year A. capillaris plants from each plot, selecting one plant from the center each of the four quadrants of the plot, at both the Tenalquot Prairie and Willow Creek sites (4 plants × 20 plots × 2 sites = 160 total plants). At the time of flowering, the plants were collected whole, dug up with the root systems intact. Shoot and root tissues were separated, and the shoot tissues were tested for Epichloë infection using the Agrinostics Field Tiller immunoblot kit (Agrinostics Ltd. Co., Watkinsville, GA, USA), and then all aboveground biomass (AGB) was dried at 60 °C for three or more days. Aboveground biomass is well established as a reliable measure of plant fitness (Shipley & Dion, 1992) and is frequently used in studies where counts of reproductive output are not feasible. In particular, it is highly correlated with reproduction in A. capillaris in our own research (Goklany, 2012).

Root tissues were cleaned and stained for quantification of percent root length colonized (PRLC) by focal symbionts. Arbuscular mycorrhizal fungi are often assessed by PRLC, providing a measure of the host/symbiont interface linked to plant fitness and phosphorus transfer (Treseder, 2013). The PRLC methods traditionally applied to AMF have only recently been applied to other root colonizing fungi, such as DSE (Weishampel & Bedford, 2006; Mandyam & Jumpponen, 2008; Upson, Read & Newsham, 2009; Dolinar & Gaberščik, 2010; Zhang, Li & Zhao, 2013). We used a modified version of Vierheilig’s ink and vinegar staining technique (Vierheilig et al., 1998), soaking roots overnight at room temperature in 10% (w/v) KOH to clear them, rinsing several times in deionized water, then staining overnight in a 5% (v/v) ink-vinegar solution using white household vinegar (5% (w/v) acetic acid) and Shaeffer’s Black drawing ink. Roots were then rinsed several times in deionized water acidified with a few drops of vinegar (Vierheilig et al., 1998). Eleven one-centimeter segments were selected at random from each root system and mounted to glass slides in polyvinyl lacto-glycerol. Slides were examined at 200× magnification and colonization percentages were obtained using McGonigle’s magnified intersections method (McGonigle et al., 1990). Arbuscules, vesicles, and hyphae were quantified. In Agrostis capillaris, we found that colonization, where present, was generally very dense, with overlapping arbuscules, vesicles, and hyphae. As such, all analyses are presented with an aggregate measure of total AMF colonization.

Statistical analyses

Soil volumetric water content was converted to soil matric potential using site-specific values of soil texture and organic matter (Saxton & Rawls, 2006), allowing for direct comparisons between sites. The average plot values of data for a twenty-day window before harvest were used in all analyses. We considered other windows, as well as temporally local maxima and minima, and found that the twenty-day window explained the most variance in the data (though 5- to 30-day windows had similar explanatory power).

Analysis of variance (ANOVA), analysis of covariance (ANCOVA), and regression analyses were used to examine effects of heating and precipitation on the fungal partners and the AGB of the host plants. We used individual plants as the replicate unit. Proportional data was transformed with the logit transformation to meet ANOVA’s requirements of normality. All ANOVA, ANCOVA, and regression analyses were performed in R version 2.15.1 (R Core Team, 2012). Site, treated as a random effect, was not significant in any model that took into account the differential N:P and soil moisture between sites, so it was excluded from analyses in favor of these variables. More extensive site characterization supports this approach to analysis (Wilson, 2012; Pfeifer-Meister et al., 2013).

Structural equation modeling (SEM), a classic multivariate technique related to multiple regression and path analysis (McCune, Grace & Urban, 2002), was used to examine hypothesized relationships among multiple symbionts within a single host, environmental conditions, and host response in the form of AGB. Given our relatively small sample size (n = 155), we attempted to meet the guideline of a 5:1 ratio of samples to free parameters (Bentler & Chou, 1987), and limited the number of selected variables within the confines of our hypothesis (Tanaka, 1987). We used bivariate scatter-plots, Pearson’s correlations, and linear regression to evaluate whether these relationships met the normality and linearity assumptions for SEM (Grace, 2006). No variables were found to possess strong co-linearity, but the soil nitrogen and phosphorus data were found individually to have almost zero explanatory power, and were thus omitted from the models. However, nitrogen-to-phosphorous ratios were kept in the models, and have been suggested by others to be a more powerful predictor of AMF responses than net availability of either nutrient alone (Johnson, 2010).

Our a priori hypotheses defined the models we tested (Fig. 2). We expected AMF and DSE to be correlated, and we expected each environmental variable (soil temperature and matric potential, as well as nitrogen-to-phosphorous ratio) to be able to affect percent root colonized by either symbiont, as well as AGB of the host. Additionally, we expected soil temperature to have a strong effect on soil matric potential, and for matric potential and temperature to have an effect on N:P ratio. We specified separate models for Epichloë-infected (E +) and Epichloë-free (E −) host plants, comparing changes in direction, magnitude, and significance of relationships to examine the effect of Epichloë infection on relationships between other symbionts, the host, and the environment. Proportional data were logit transformed to satisfy distributional and linearity assumptions. Plant was again used as the unit of replication.

Figure 2 Structural equation model.

Schematic of our Structural Equation Model, which illustrates our a priori hypotheses. Arrows represent predicted direct effects of one variable on another; double headed arrows represent correlations.

The relationships amongst all variables were modeled as path coefficients, which represent the magnitude and direction of the effect of each predictor variable on a response variable with all other variables held constant. SEM analysis was conducted in IBM’s SPSS Amos (v20.0) software (SPSS Inc., Chicago IL, USA), using a maximum likelihood approach to model evaluation and parameter estimation. Model fit was evaluated using the χ2 goodness-of-fit statistic and associated p-values, Bentler Comparative Fit Index (CFI), and Root Mean Square Error of Approximation (RMSEA). CFI values range between 0 and 1, with higher values indicating better model fit (Bentler & Chou, 1987), and tend to underestimate model fit when sample sizes are small (Bishop & Schemske, 1998).

Results

For ease of comparison throughout this paper, figure-elements representing groups/samples hosting Epichloë endophytes are shown in red (E + ), while those groups/samples not hosting Epichloë endophytes are shown in blue (E − ).

Infection with Epichloë was 36% (n = 155), and was uncorrelated with any environmental variable (see Figs. S1–S11). We found no evidence of competition between symbionts. Epichloë infection did not affect root length colonized by either AMF (Fig. 3A; F1,153 = 0.956, P = 0.330) or DSE (Fig. 3B; F1,153 = 0.083, P = 0.774). Percent root length colonized by AMF and DSE were correlated positively (Fig. 4 and Fig. S1; Adjusted R2 = 0.107, F1,153 = 19.51, P < 0.001), indicating facilitation rather than competition.

Figure 3 PRLC.

Percent root length colonized by AMF (A) and DSE (B) for plants without Epichloë endophytes (E −, blue), and those hosting Epichloë endophytes (E +, red).

Structural equation model fit was good; both the E + (n = 56) and E − (n = 99) structural equation models had non-significant χ2 values (P > 0.10) and Bentler CFIs > 0.90. Magnitude of standardized path coefficients differed between the E + and E − models, but these were relatively minor. The only substantial difference between the models was a negative correlation between DSE root length colonized and plant biomass, but only in the absence of Epichloë infection (Fig. 4 and Fig. S2; E + Adjusted R2 = 0.029, F1,54 = 2.644, P = 0.110; E − Adjusted R2 = 0.053, F1,97 = 6.437, P = 0.013). DSE colonization decreased when more water was available to plants (Fig. 4 and Fig. S3; Adjusted R2 = 0.107, F1,153 = 19.5, P < 0.001). There was a direct negative effect of warmer soil temperatures on DSE colonization as well, which regression analysis does not recover (Fig. 4). Neither AMF colonization nor proportion of plants hosting Epichloë varied significantly with measured edaphic conditions (soil moisture, soil temperature, soil N:P ratios; Fig. 4 and Figs. S4–S9).

Figure 4 Structural equation model results.

Overall SEMs, with different models for those plants without Epichloë endophytes (A: E −, blue), and those with Epichloë endophytes (B: E +, red). Model fit was good for both models (A: χ2 = 2.50, P = 0.114; CFI = 0.981; RMSEA = 0.124; n = 99|B: χ2 = 0.63, P = 0.427; CFI = 1.000; RMSEA < 0.001; n = 56). The numbers above the arrows are the standardized path coefficients. Non-significant (P > 0.05) path coefficients are not shown. Numbers in the boxes are total explained variance (R2) of each variable.

Discussion

Our initial hypotheses centered on competition between symbionts within a shared host: we expected to find evidence that consortia of symbionts changed with environmental conditions in such a way as to minimize changes to host fitness (and maximize fitness in a given environmental context). In other words, we expected there to be interactions among environmental variables (soil temperature, moisture, and N:P ratios) and the fitness costs/benefits of colonization by different symbionts. In addition, we expected to find evidence of competition between symbionts. Lastly, we expected the outcomes of that competition to be stabilized by the fitness benefits to the host.

What we found instead was no evidence of competition between symbionts: neither root symbiont seems to be affected by presence of Epichloë endophytes in the aboveground tissues of the host (Fig. 3). If anything, AMF and DSE appeared to have a facilitative rather than a competitive interaction (Fig. 4 and Fig. S1). We also did not find any effect of AMF colonization or Epichloë presence on plant fitness as measured by AGB (Fig. 4 and Figs. S10–S11). Aboveground biomass is known to be highly correlated with reproduction in A. capillaris (Goklany, 2012), and is often used as a surrogate for overall fitness (Shipley & Dion, 1992). We did find a negative effect of DSE colonization on AGB, but only in the absence of Epichloë endophytes (Fig. 4 and Fig. S2), suggesting that the presence of Epichloë counteracts the otherwise negative effects of DSE.

DSE/Epichloë interaction

To our knowledge, this is the first study to examine interactions between DSE and Epichloë endophytes. We found a significant effect of DSE root length colonized on plant biomass, but only when the host plants did not also host foliar Epichloë endophytes.

Dark septate root endophytes have been studied very little, though there has been broader interest recently (Porras-Alfaro & Bayman, 2011; Mandyam & Jumpponen, 2015). These fungi show a wide range of effects on their host plants, from mutualism to pathogenicity (Jumpponen, 2001; Mandyam & Jumpponen, 2005; Grünig et al., 2008; Alberton, Kuyper & Summerbell, 2010; Newsham, 2011; Mandyam, Fox & Jumpponen, 2012; Mandyam, Roe & Jumpponen, 2013; Mayerhofer, Kernaghan & Harper, 2013). The variability of response of the host plant is likely linked to the variability of the DSE species being studied, the genetic combinations of particular host/symbiont pairs, and the environmental context within which the experiment takes place (Mandyam, Fox & Jumpponen, 2012; Mandyam & Jumpponen, 2015). That environmental context includes the entire consortium of interacting fungal symbionts within a given host (Munkvold et al., 2004; Grünig et al., 2008; Mandyam, Fox & Jumpponen, 2012; Mandyam, Roe & Jumpponen, 2013), as our findings demonstrate.

Inoculation studies support the function of DSE as ‘pseudo-mycorrhizal’ in that they have been shown to translocate N or P into their hosts (Jumpponen, Mattson & Trappe, 1998; Newsham, 2011), but N uptake seems to be the more common role for DSE in this context (Upson, Read & Newsham, 2009; Alberton, Kuyper & Summerbell, 2010; Newsham, 2011). Epichloë endophytes are well known for producing fungal alkaloids which discourage herbivory (Schardl, 1996; Bush, Wilkinson & Schardl, 1997), including those species known to associate with Agrostis capillaris (Funk, White & Breen, 1993; Porter, 1995; Schardl & Phillips, 1997; Leuchtmann, Schmidt & Bush, 2000). These alkaloids are costly to produce, particularly in terms of nitrogen (Belesky et al., 1988; Faeth & Fagan, 2002)—although, it has been suggested that carbon may also limit alkaloid biosynthesis (Rasmussen et al., 2008). Thus, herbivory reduction by Epichloë endophytes may be dependent upon soil nutrient levels (Lehtonen, Helander & Saikkonen, 2004). We theorize that the interaction we saw between DSE root length colonized and Epichloë infection may be the intersection of these two things: in the absence of an Epichloë infection, the fitness increase from N gained by hosting more DSE is not offset by the metabolic (i.e., carbon) cost of hosting the DSE, but when also hosting Epichloë endophytes, the increased N uptake can be allocated to plant defense by way of fungal alkaloids, thus offsetting the costs of hosting the DSE. Such interactions may also be affected by priority effects, and may be differential in the case of seedborne Epichloë transmission versus horizontal transmission. Much more work will be required to investigate this theory.

AMF/DSE interaction

We found a positive correlation between AMF root length colonized and DSE root length colonized, as well as generally high colonization values for both fungi (Figs. 3, 4 and Fig. S1). This correlation was not influenced by Epichloë endophyte infection, site, or climate treatment.

The few studies to date examining the interactions between these two common root symbionts have found conflicting results. Competition between AMF and DSE is reported from wetland plants in Louisiana by Kandalepas and colleagues (2010), who found that plants that had greater AMF colonization generally had lower DSE colonization, and vice versa. These results are similar to those of Urcelay and colleagues (2011), who report that high alpine species of the Altiplano in Bolivia display evidence of a tradeoff between AMF and DSE root colonization. However, a study in Chinese grasslands found that DSE colonization was generally positively correlated with AMF hyphal—but not arbuscular or vesicular—colonization (Lingfei, Anna & Zhiwei, 2005).

However, these studies examined variation in AMF/DSE colonization between host species, not within a single host species. Within the bounds of variation for a particular host species, the relationship between the two symbionts might be quite different; for example, Scervino and colleagues (2009) found that exudates from a particular DSE could stimulate lengthening and branching of AMF hyphae in vitro, which indicates a facilitatory effect, consistent with our results; interestingly, similar effects have been observed with exudates from Epichloë endophytes (Novas et al., 2011). Future research should focus on these host/symbiont pair-specific interactions within single plant host species.

Context-dependence

Given the broad importance of AMF, DSE, and Epichloë symbioses to ecological (Porras-Alfaro & Bayman, 2011; Mohan et al., 2014; Mandyam & Jumpponen, 2015) and economic systems (Hoveland, 1993; Dodd, 2000), we feel it is important to emphasize that the system of interaction we have observed here represents a single set of symbioses. As discussed above, the identities and genetic backgrounds of the particular host/symbiont partners are of great importance to the outcome of the association (Ahlholm et al., 2002; Klironomos, 2003; Mandyam & Jumpponen, 2015); additionally, the environmental context within which a particular host/symbiont pair interact is of great importance to the outcome of the association (Ahlholm et al., 2002; Landis, Gargas & Givnish, 2004; Roy, Güsewell & Harte, 2004; Mandyam & Jumpponen, 2015).

In an attempt to examine the generalizability of these results, we initiated a small, similar study, also within the context of the larger manipulative climate change experiment (data available in Vandegrift et al., 2015b). We used the annual grass Bromus hordeaceus L. for this experiment, and collected data in a similar manner, but only at the southern-most site, which has much greater soil nutrient availability and total precipitation, but also much more extreme seasonal climate variation (see Pfeifer-Meister et al., 2013). These samples from only a single site covered a much narrower climatic envelope than the Agrostis dataset, and were much more limited in sample size, particularly the E + samples (n = 19). With these caveats in mind, we found very different results: in the Bromus dataset, Epichloë infection changed the response of AMF, DSE, and host AGB to environmental variables; there was no correlation between AMF and DSE; and while Epichloë infection still modulated the effect of DSE colonization, the effect of DSE colonization on E − plants was positive, not negative (Fig. S12).

These differences highlight that the spectrum of host responses to symbiont consortia and environmental conditions is very much dependent upon the identities of the host and symbionts, as well as the particular set of environmental conditions within which the host/symbiont groupings are set. The importance of context-dependence, and species-specific idiosyncratic responses to abiotic factors has long been noted (Brown & Ewel, 1987; Wardle et al., 2004; Roy, Güsewell & Harte, 2004; Agrawal et al., 2007).

This study relies on microscopic observation of DSE and AMF, and immunoblot identification of Epichloë infections, which limits our ability to determine specific species-by-species interactions between the symbionts. As discussed in the introduction, DSE are very phylogenetically diverse (Porras-Alfaro & Bayman, 2011), and though they are often treated as a single functional group, they may play very different roles in different contexts simply because they are different organisms (Mandyam & Jumpponen, 2015). Similarly, different species of AMF have been shown to have different functional roles (Munkvold et al., 2004), which may interact differently with DSE and Epichloë symbionts. Future work should focus on connecting the functional roles of these various symbionts with particular taxonomic groups, and attempt to link fungal microbiome data with careful microscopic observation across climatic gradients.

Integration of effects of symbiont consortia

Given the preponderance of emerging data about the complexity of AMF, DSE, and Epichloë endophyte ecology, a conceptual framework that synthesizes these advances is clearly necessary. Such a conceptual framework must take into account evidence for all partners, including: host specificity (Leuchtmann, 1993; Vandenkoornhuyse et al., 2003; Martínez-García & Pugnaire, 2011) and host generalism (Bever et al., 2001; Grünig et al., 2008; Smith & Read, 2008); the functional diversity of fungal partners, even within single functional groups like AMF (Helgason et al., 2002; Öpik et al., 2009); colonization of the same host individual by multiple species of fungi (Palmer et al., 2010; Mandyam & Jumpponen, 2015); changes in symbiont communities with changes in the abiotic environment (Martínez-García & Pugnaire, 2011), including seasonal changes (Bever et al., 2001); and the co-evolutionary history between terrestrial plants and their mycobiota (Carroll, 1988).

Facilitation between fungal species within a host may play a role in symbiont community determination: it has been demonstrated that both DSE and Epichloë derived exudates can affect the growth of AMF (Scervino et al., 2009; Novas et al., 2011), and our study supports facilitation between AMF and DSE, as well as synergistic effect of DSE colonization and Epichloë infection on host fitness. Indeed, facilitatory interactions need not be restricted to within single hosts: given the demonstrated movement of photosynthate between host species through mycorrhizal networks (Martins & Read, 1996; Martins & Cruz, 1998; Pringle, 2009), connectivity between hosts by different species of fungi may be just as important to supporting struggling populations of fungi as it is to struggling plants.

Given this community framework, it is reasonable to expect that selective pressure on the host will favor host/symbiont relationships that structure the community of symbionts in the most beneficial way possible for the plant, not necessarily the individual symbiont that is most beneficial to plant fitness in isolation. The fitness effect of the consortium of symbionts is the integration of all fitness costs and benefits of all partners. The particular community assemblage of symbiotic fungi associated with a particular host will then be predicated upon the physiology of the host, the available inoculum, the interactions of the symbionts, and the abiotic environment’s effects on both the host and the fungal partners (Schlaeppi & Bulgarelli, 2015).

Supplemental Information

Supplemental Information 1 This is the collection of supplemental figures referenced in the text of the article, with associated legends

Click here for additional data file.

We thank The Nature Conservancy (TNC) and the Center for Natural Lands Management (CNLM) for site use. Maya Goklany, Lorien Reynolds, Timothy Tomaszewski, and Hannah Wilson all provided invaluable assistance with collection, processing, and data analysis. Kelly Campbell, Amanda Clark, Matthew Davis, and Ashley Ludden provided tireless help in the lab.

Additional Information and Declarations

Competing Interests

Author Contributions

Field Study Permissions

Data Availability

The authors declare there are no competing interests.

Roo Vandegrift conceived and designed the experiments, performed the experiments, analyzed the data, wrote the paper, prepared figures and/or tables, reviewed drafts of the paper.

Bitty A. Roy and Scott D. Bridgham conceived and designed the experiments, contributed reagents/materials/analysis tools, reviewed drafts of the paper.

Laurel Pfeifer-Meister conceived and designed the experiments, performed the experiments, reviewed drafts of the paper.

Bart R. Johnson conceived and designed the experiments, reviewed drafts of the paper.

The following information was supplied relating to field study approvals (i.e., approving body and any reference numbers):

We obtained memoranda of understanding with both The Nature Conservancy (TNC) and the Center for Natural Lands Management (CNLM) for conducting field experiments on their lands. These MOUs are confidential.

The following information was supplied regarding data availability:

Figshare:

http://dx.doi.org/10.6084/m9.figshare.1453199 and

http://figshare.com/articles/Data_from_The_herbaceous_landlord_symbiont_consortia_lead_to_context_dependent_interactions_within_a_single_host/1453199.

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
