# Peer review of "The herbaceous landlord: integrating the effects of symbiont consortia within a single host"

_PeerJ, doi:10.7717/peerj.1379_

## Round 0.1 · original submission · Major Revisions

Both reviewers found your study interesting and topical.  It provides new insights into the interaction between Epichloë endophytes, arbuscular mycorrhizae, and DSE.  Nonetheless, a few concerns and methodological and conceptual issues were raised. I have noted the following:

- The description of the experimental design should be improved. Some sections are hard to grasp (see reviewer #1's comments);
- Please use PCA and linear discrimination or canonical correlation to tease out other correlations (see reviewer #2's comments);
- The box plots in the main figures display negative information and could have been presented in a more concise format such as a table (see reviewer #2's comments).
- Discussion on positive and negative interactions under different climate conditions should be reinforced (both reviewers).

I think reviewers raised some good points here that you need to take into account. As both reviewers, I am mostly concern by the fact that no attempts were made to identify the species of fungi that became associated with the roots. This is a clear limitation of the present study. A molecular description of the microbiome would have been a nice addition to the paper. Is this information available? If so, it should be included in this paper. At the least, this issue should have been given some attention in the discussion.

Reviewer 1 ·

Basic reporting

The article is well written. The title is excellent. A couple of points need to be mentioned in the background:
The Epichloë fungi are seed-transmitted and therefore this endophyte will be in the seedlings before the root endophytes come into contact with the plant. This should be mentioned in the introduction and discussion.
The authors mention that it is logical that the endophytes will interact with each other, yet do not mention that Epichloë fungi are not generally found in the roots, or if present, only in the root meristem of the primary root. The bulk of the hyphae are in the shoot apical meristem and other aerial parts of the grass. Any interactions between the endophytes will therefore be indirect and predominantly through signalling. This should also be mentioned.

Experimental design

I had some difficulty in interpreting how the Agrostis capillaris plants got into the test plots in the first place. The authors mention glyphosate-treating the plots twice, and then seeding them with a mixture of 32 “native upland prairie graminoids and forbs”. After that, they describe Agrostis capillaris as an introduced (i.e. not native) species. Were the test plants germinated in a glasshouse and then transplanted into the plots? If germinated in potting mix in another environment, is it possible they were exposed to root endophytes prior to going into the field? Alternatively, were the A. capillaris seeds included in the seed mix? If so, this should be mentioned, and then how did the authors select the 4 plants from each plot for endophyte analysis? Were quadrants thrown to ensure there was no bias?
No attempts were made to identify the species of fungi that became associated with the roots. Dark septate fungi are highly diverse, being found in different phyla, my question is whether they may have identified correlations between the different root endophytes if species identification had been attempted. At the least, this issue should have been given some attention in the discussion.

Validity of the findings

The experiments had plenty of replication. The statistics used were univariate and therefore not particularly illuminating. The authors may want to consider using PCA and linear discrimination or canonical correlation to tease out other correlations. The box plots in the main figures presented negative information and could have been presented in a more concise format such as a table. This also applies to the figures in the supplementary data which mostly show no correlations and which should be presented as tables. I am not familiar with structural equation modelling and therefore not able to comment on whether this was an appropriate tool to use, and whether the data met the appropriate criteria for its use.

Additional comments

No attempts were made to identify the species of fungi that became associated with the roots. Dark septate fungi are highly diverse, being found in different phyla, my question is whether they may have identified correlations between the different root endophytes if species identification had been attempted. At the least, this issue should have been given some attention in the discussion.

Further observations:
Line 27: add (PNW) after Pacific North West as you use the abbreviation later.
Line 144: alter “well know” to “well known”
Line 164: Add “are” after endophyte
Line 202: 7.1 m2 area or 7.1 m diameter?
Line 202: change “each sites” to “each site”
Line 224: Add citation
Line 247: wt/vol should read w/v and be placed after the other % in the same paragraph
Line 250: transfer “several times” to after “rinsed” otherwise this implies that you acidified the water several times.
Line 271: In the sentence starting “Site…” the tense is different to the rest of the methods.
Line 487: Barry, S. should read Scott, B. This needs to be changed in the text as well.

Reviewer 2 ·

Basic reporting

Very little is known about the nature of the association in terms of the microbiome. The authors proposed a study utilizing one species of grass host infected or not with Epichloë endophytes, arbuscular mycorrhizae, and DSE. The work' major objective was to evaluate how the relationships among three different symbionts and their host change across environmental conditions. To accomplish the objective, the authors performed a climate manipulative experiment. Based on the results, the authors suggest that selective pressure on the host is likely to favor host/symbiont relationships that structure the community of symbionts in the most beneficial way possible for the host, not necessarily favoring the individual symbiont that is most beneficial to the host in isolation.
The work seems to be well conducted and the overall quality of the text is good. However, I consider that the results need additional analyses to improve the discussion:
• Do the authors know the identity of the Epichloë species associated with the host? This information would be quite interesting to improve the Discussion.
• Interactions among symbionts and host is interesting describe by the authors, but a little vague. Discussion on positive and negative interactions under different climate conditions should be reinforced. Mechanisms involved?

Experimental design

• It is no clear from this section how many E+ and E- plants were analyzed.
Line. 254. Authors explained that they “found that colonization, where present, was generally very dense, with overlapping arbuscules, vesicles, and hyphae”, so results “are present with an aggregate measure of total AMF colonization”. However the McGonigle’s magnified intersections method allows to discriminate and quantify precisely each type of AMF structure at each intersection point. I believe it would be interesting if the authors would include the data discriminate by structure type, thus more information may be obtained from the experiment performed.
Line 292. If you collected plants at random at each site, how do you get an equal or at least similar ratio of E+ and E- plants.

Validity of the findings

No Comments

Additional comments

Lines 370-376. Do the authors have any record of the production of alkaloids by Agrostis capillaris? If not this paragraph is too much speculative.
Figure 2. Authors should detail in the captions the abbreviations used in the figure.

---

## Round 0.2 · accepted · Accept

I hope that future research will allow you to study the Epichloe-DSE-AMF interactions at the molecular level.